# Safety-critical Obstacle Avoidance Control of Autonomous Surface Vehicles with Uncertainties and Disturbances

1st Gege Dong
*College of Marine Electrical Engineering*
*Dalian Maritime University*
Dalian, China
donggege0507@163.com

2nd Li-Ying Hao*
*College of Marine Electrical Engineering*
*Dalian Maritime University*
Dalian, China
haoliying_0305@163.com

*Abstract*—This paper proposes a safety-critical obstacle avoidance control approach for autonomous surface vehicles (ASVs) with disturbances and uncertainties. The existing exponential control barrier functions (ECBF) are extended to handle unknown disturbances, leading to the development of input-to-state safe exponential control barrier functions (ISSf-ECBFs). An extended state observer is used to estimate unknown external marine disturbances and internal model uncertainties, based on which an anti-disturbance controller is designed. Based on the proposed ISSf-ECBFs, a quadratic programming problem is formulated to determine the optimal control input. It is proven that the closed-loop system is input-to-state safe and the errors of the closed-loop system are uniformly ultimately bounded. Simulations validate the effectiveness of the proposed control strategy.

*Index Terms*—Autonomous surface vehicles (ASVs), safety-critical control, obstacles avoidance, input-to-state safe exponential control barrier functions (ISSf-ECBFs)

## I. INTRODUCTION

Autonomous Surface Vehicles (ASVs) are gaining attention for their ability to enhance maritime operations [1]–[3]. With advanced sensors and navigation systems, ASVs can navigate complex environments and perform diverse tasks [4]. They are increasingly utilized in search and rescue, fisheries management, hydrographic surveying, and offshore energy, making them a focal point for researchers in ASV control [5]–[7].

ASVs navigating in dynamic marine environments face numerous challenges, primarily internal model uncertainties and external disturbances [8]. Internal uncertainties arise from modeling inaccuracies, parameter variations, and sensor noise. Additionally, ASVs must navigate unpredictable ocean conditions, such as waves, currents, and winds. These factors can adversely affect the performance of control strategy. To address this challenge, researchers have proposed various methods to enhance the robustness of the system, such as sliding mode control [9], adaptive control, and neural network control [10].

This work was funded by the National Natural Science Foundation of China (51939001, 52171292, 51979020, 61976033), Dalian Outstanding Young Talents Program (2022RJ05), the Topnotch Young Talents Program of China (36261402), and the Liaoning Revitalization Talents Program (XLYC2007188).

The Extended State Observer (ESO) can estimate disturbances in real-time and dynamically adjust the control strategy. By treating internal model uncertainties and external disturbances as lumped disturbances for estimation, the reliance on the model can be reduced, thereby enhancing the robustness of the system.

In complex maritime environments, ASVs face significant threats from various obstacles, including vessels, islands, and reefs [11]. To mitigate these risks, researchers have proposed several obstacle avoidance strategies, such as the artificial potential method [12], the velocity obstacle method [13], and the dynamic window approach [14]. Control barrier functions (CBFs), introduced in [15], have proven effective in ensuring real-time safety. In [16], the nominal controller was modified to formally adhere to safety constraints for successful obstacle avoidance. However, the control strategy in [16] did not account for model uncertainties or disturbances. To address this, [17] introduced input-to-state safe control barrier functions (ISSf-CBFs). Furthermore, [18] proposed a framework to ensure safety for uncertain nonlinear systems with structured parametric uncertainty. In [19], a collision avoidance strategy for ASVs was proposed using ISSf-CBFs. However, these functions have a relative degree of one, limiting their use in higher-order systems. To address this, [20] introduced exponential control barrier functions (ECBFs). [21] further explored ISSf-ECBFs under known perturbation bounds, but measuring such disturbances is challenging. Therefore, a safety-critical controller based on ISSf-ECBFs is crucial for ASVs dealing with unknown model uncertainties and external disturbances.

This paper presents a safety-critical control strategy for Autonomous Surface Vehicles (ASVs) that accounts for external marine disturbances and internal model uncertainties. The key contributions are as follows:

1) While the existing method [21] constructs safety constraints only under known disturbances or their upper bounds, this paper extends the results of input-to-state safe control barrier functions (ISSf-ECBFs) to develop safety constraints for unknown disturbances.

2) Unlike previous work [12], [22]–[24], this paper formulates a safety-critical controller based on ISSf-ECBFs by constructing a quadratic programming problem to facilitate collision avoidance with obstacles.

The structure of the paper includes the following sections: The preliminaries and problem statement are covered in Section II. Section III gives the safety-critical controller design and Section IV is stability and safety analysis. Simulations are carried out in Section V. Section VI summarizes this article.

## II. PRELIMINARIES AND PROBLEM STATEMENT

### A. Notation

In this paper, the notation $\|\cdot\|$ denotes the 2-norm of a vector, and $\Re$ represents the set of real numbers. The symbols $\lambda_{\min}(\cdot)$ and $\lambda_{\max}(\cdot)$ indicate the smallest and largest eigenvalues of a symmetric matrix, respectively.

Let $\beta(r)$ be a scalar continuous function defined for $r \in [-b, a)$. If $a = \infty$, $b = 0$, and $\beta(r) \to \infty$ as $r \to \infty$, then $\beta(r)$ belongs to class $\mathcal{K}_\infty$. If $a, b = \infty$, $\beta(r) \to \infty$ as $r \to \infty$, and $\beta(r) \to -\infty$ as $r \to -\infty$, it represents an extended class $\mathcal{K}_\infty$, denoted as $\mathcal{K}_{\infty,e}$.

### B. Input-to-state Safe Exponential Control Barrier Functions

Consider the following system

$$\dot{x} = f(x) + g(x)u + d_w \quad (1)$$

where $x(t) \in \Re^n$ denotes state and $u \in \Re^m$ denotes control input. The term $d_w$ denotes bounded disturbances. The function $f(x) \in \Re^n$ and $g(x) \in \Re^{n \times m}$ are locally Lipschitz continuous.

**Definition 1.** *[25] The set $\mathcal{C} \in \Re^n$ is described as*

$$\mathcal{C} \triangleq \{x \in \Re^n \mid S(x) \geq 0\}$$
$$\partial\mathcal{C} \triangleq \{x \in \Re^n \mid S(x) = 0\}$$
$$Int(\mathcal{C}) \triangleq \{x \in \Re^n \mid S(x) > 0\} \quad (2)$$

*where $h(\cdot) \in \Re^n \mapsto \Re$ represents a continuously differentiable function, and $\mathcal{C}$ is referred to as the safe set. If for all $x_0 \in \mathcal{C}$ it holds that $x(t) \in \mathcal{C}$ for every $t \in I(x_0)$, then the set $\mathcal{C}$ is considered forward invariant. Consequently, the system described by (1) with $d_w(t) = 0$ can be deemed safe on $\mathcal{C}$.*

**Definition 2.** *The relative degree of $S(x) : \Re^n \to \Re$ with respect to the system (1) refers to the number of derivatives required along the dynamics of (1) before the control input $u$ explicitly appears.*

**Definition 3.** *[17] For system (1), an extended set $\mathcal{C}_d \supset \mathcal{C}$ is expressed as follows*

$$\mathcal{C}_d \triangleq \{x \in \Re^n \mid S(x) + \beta_d(\|d_w(t)\|_\infty) \geq 0\}$$
$$\partial\mathcal{C}_d \triangleq \{x \in \Re^n \mid S(x) + \beta_d(\|d_w(t)\|_\infty) = 0\}$$
$$Int(\mathcal{C}_d) \triangleq \{x \in \Re^n \mid S(x) + \beta_d(\|d_w(t)\|_\infty) > 0\} \quad (3)$$

*where $\|d_w\|_\infty \leq \bar{d}_w$, a positive constant, and $S(x)$ is a continuous function, with $\beta_d(\cdot) \in \mathcal{K}_{\infty,e}$.*

**Definition 4.** *(ISSf [17]) If the control input $u$ and the function $\beta_d$ ensure the forward invariance of the set $\mathcal{C}_d$, then the system (1) with disturbances is ISSf on $\mathcal{C}$.*

**Definition 5.** *(ISSf-ECBF [17]) Considering the sets $\mathcal{C}_d$ defined by (3), $S(x)$, which has a relative degree $\rho > 1$, qualifies as an ISSf-ECBF for the system described in (1). This holds true if, for all $x \in \Re^n$, there exist a bound $\|d_w\|_\infty \leq \bar{\tau}_w$ and a function $\gamma(\cdot) \in \mathcal{K}_{\infty,e}$ that satisfies*

$$\sup_{u \in \mathcal{U}}[\mathcal{L}_f^\rho S(x) + \mathcal{L}_g\mathcal{L}_f^{\rho-1}S(x)u + (\frac{\partial(\mathcal{L}_f^{\rho-1}S(x))}{\partial x})^T d_w$$
$$+ \mathcal{T}_s^T \mathcal{H}_s] \geq -\gamma(\|d_w\|_\infty) \quad (4)$$

*The terms $\mathcal{L}_f^\rho$ and $\mathcal{L}_g\mathcal{L}_f^{\rho-1}$ represent the Lie derivatives of the function $S(x)$. $\mathcal{T}_s = \begin{bmatrix} p_0 & p_1 & \dots & p_\iota \end{bmatrix}^T$ where $p_i$ is positive constant. $\mathcal{H}_s = [S(x) \quad \mathcal{L}_f S(x) \quad \dots \quad \mathcal{L}_f^{\rho-1} S(x)]^T$.*

**Lemma 1.** *If $S(x)$ functions as an ISSf-ECBF for the system (1) in the set $\mathcal{C}$, then any controller $u \in \mathcal{U}$ that is Lipschitz continuous and valid for all $x \in \Re^n$ must satisfy*

$$\mathcal{U}(x) = \{u \in \Re^m : \mathcal{L}_f^\rho S(x) + \mathcal{L}_g\mathcal{L}_f^{\rho-1}S(x)u$$
$$+ (\frac{\partial(\mathcal{L}_f^{\rho-1}S(x))}{\partial x})^T d_w + \mathcal{T}_s^T \mathcal{H}_s \geq -\gamma(\|d_w\|_\infty)\}. \quad (5)$$

*This implies that the set $\mathcal{C}_d$ is forward invariant. In other words, the system (1) is ISSf on the set $\mathcal{C}$.*

### C. ASV Model

The kinematics and kinetics of ASV can be described as:

$$\dot{\eta}(t) = R(\psi)\nu(t)$$
$$M\dot{\nu}(t) = f(\nu) + d_w(t) + \tau(t) \quad (6)$$

where $\eta(t) = \begin{bmatrix} \bar{p}(t) & \psi(t) \end{bmatrix}^T \in \Re^3$ represents the position and heading of ASV. $R(\psi) = \text{diag}\{R_2(\psi), 1\}$ is a rotate matrix with

$$R_2(\psi) = \begin{bmatrix} \cos(\psi) & -\sin(\psi) \\ \sin(\psi) & \cos(\psi) \end{bmatrix}. \quad (7)$$

The vector $\nu(t) = \begin{bmatrix} u(t) & v(t) & r(t) \end{bmatrix}^T \in \Re^3$ represents the surge velocity, sway velocity, and yaw velocity, respectively. The matrix $M$ denotes the inertial matrix. $f(\nu)$ represents the Coriolis and centripetal matrix, damping matrix, and unmodeled hydrodynamics. The vector $\tau(t)$ signifies the forces produced by the actuators. The external disturbances, caused by wind, waves, and ocean currents, are represented by $d_w(t) = \begin{bmatrix} d_{w1}(t) & d_{w2}(t) & d_{w3}(t) \end{bmatrix}^T \in \Re^3$.

Letting $q(t) = R(\psi)\nu(t)$, (6) can be rewritten as

$$\dot{p} = q$$
$$\dot{q} = \xi + RM^{-1}\tau \quad (8)$$

where $\xi = RM^{-1}(d_w + f(\nu)) + \dot{R}\nu$.

The desired parameterized path is set as $p_0(\theta) = [x_0(\theta), y_0(\theta), \psi_0(\theta)]^T$, $\psi_0(\theta) = \arctan(y_0^\theta(\theta)/x_0^\theta(\theta))$ where $\theta$ represents path variable. $y_0^\theta(\theta)$ and $x_0^\theta(\theta)$ is the partial derivative of $y_0(\theta)$ and $x_0(\theta)$, respectively. In addition, it is assumed that $p_0^\theta(\theta)$ is bounded.

### D. Problem Formulation

The safety-critical obstacle avoidance controller of ASV is required to achieve the following tasks:

(1) Geometric task: Ensure that the ASV follows the desired path, meaning that

$$\lim_{t\to\infty} \|p(t) - p_0(\theta)\| < l_1 \qquad (9)$$

where $l_1 \in \mathfrak{R}$ denotes a small positive constant.

(2) Dynamic task: The derivative of the path variable $\theta$ converge to the desired speed

$$\lim_{t\to\infty} \|\dot{\theta}(t) - u_d(t)\| < l_2 \qquad (10)$$

where $u_d(t)$ represents desired speed and $l_2$ is a small positive constant.

(3) Obstacle avoidance task: To prevent collisions between the ASV and obstacles, the following condition must be met

$$\|\bar{p}(t) - \bar{p}_k(t)\| > r_k + d_k \qquad (11)$$

where $\bar{p}_k(t)$, $r_k$, and $d_k$ represent the position, the radius, and the minimum obstacle avoidance distance of the $k$th obstacle.

## III. MAIN RESULTS

### A. ISSf-ECBF with Unknown Disturbances

While the previous studies have made substantial progress, they were mainly directed at scenarios with known disturbances or predefined upper bounds. To alleviate this limitation, the following theorem is presented to account for unknown disturbances.

**Theorem 1.** *Given the ISSf-ECBF $S(x)$ as defined in Definition 5 for the system (1) on the set $\mathcal{C}$, if there exists a bound $\|d_w\|_\infty \leq \bar{d}_w$ such that for every $x \in \mathfrak{R}^n$, the following inequality holds*

$$\sup_{u\in\mathcal{U}}[\mathcal{L}_f^\rho S(x) + \mathcal{L}_g\mathcal{L}_f^{\rho-1}S(x)u + \mathcal{T}_s^T\mathcal{H}_s$$
$$- (\frac{\partial(\mathcal{L}_f^{\rho-1}S(x))}{\partial x})^T(\frac{\partial(\mathcal{L}_f^{\rho-1}S(x))}{\partial x})] \geq 0 \quad (12)$$

*and the admissible control set satisfies as*

$$\mathcal{U}(x) = \{u \in \mathfrak{R}^m : \mathcal{L}_f^\rho S(x) + \mathcal{L}_g\mathcal{L}_f^{\rho-1}S(x)u + \mathcal{T}_s^T\mathcal{H}_s$$
$$- (\frac{\partial(\mathcal{L}_f^{\rho-1}S(x))}{\partial x})^T(\frac{\partial(\mathcal{L}_f^{\rho-1}S(x))}{\partial x}) \geq 0\}. \quad (13)$$

*Then, we can obtain that the system (1) is ISSf on $\mathcal{C}$.*

*Proof.* For $u \in \mathcal{U}(x)$, one has

$$\mathcal{L}_f^\rho S(x) + \mathcal{L}_g\mathcal{L}_f^{\rho-1}S(x)u + (\frac{\partial(\mathcal{L}_f^{\rho-1}S(x))}{\partial x})^T d_w + \mathcal{T}_s^T\mathcal{H}_s$$
$$\geq (\frac{\partial(\mathcal{L}_f^{\rho-1}S(x))}{\partial x})^T(\frac{\partial(\mathcal{L}_f^{\rho-1}S(x))}{\partial x}) + (\frac{\partial(\mathcal{L}_f^{\rho-1}S(x))}{\partial x})^T d_w$$
$$\geq (\frac{\partial(\mathcal{L}_f^{\rho-1}S(x))}{\partial x})^T(\frac{\partial(\mathcal{L}_f^{\rho-1}S(x))}{\partial x})$$
$$- \|\frac{\partial(\mathcal{L}_f^{\rho-1}S(x))}{\partial x}\|\|d_w\|_\infty. \quad (14)$$

Adding and subtracting $\frac{\|d_w\|_\infty^2}{4}$ yields

$$\dot{h} \geq (\frac{\partial(\mathcal{L}_f^{\rho-1}S(x))}{\partial x} - \frac{\|d_w\|_\infty}{2})^2 - \frac{\|d_w\|_\infty^2}{4}$$
$$\geq -\frac{\|d_w\|_\infty^2}{4} \qquad (15)$$

which is of the form (4). $\qquad\square$

**Remark 1.** *Compared with [21], the proposed ISSf-ECBF can deal with unknown perturbations. Although asymptotically stable ESO is used in reference 21, in practice, the disturbance estimation error is difficult to be 0. Thus, it is essential to develop ISSf-ECBFs that ensure safety in the presence of unknown disturbances.*

### B. Anti-disturbance Controller Design

In this section, we will focus on designing a safety-critical controller. The control architecture for the proposed strategy is illustrated in Figure 1.

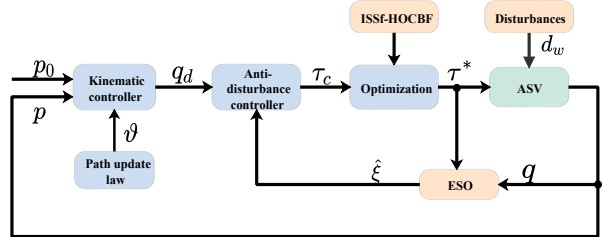

Fig. 1. Control architecture of the safety-critical controller for the ASV.

Firstly, we utilize the ESO to obtain the estimations of the model uncertainties, external disturbances in this part. In addition, the ESO relies on the following assumption.

**Assumption 1.** *$\dot{\xi}(t)$ is a bounded function meeting*

$$\|\dot{\xi}(t)\| \leq \xi^* \qquad (16)$$

*where $\xi^*$ is a positive constant.*

Then, the ESO is devised to estimate model uncertainties, external disturbances.

$$\begin{cases} \dot{\hat{q}}(t) = -K_1\tilde{q}(t) + \hat{\xi}(t) + RM^{-1}\tau \\ \dot{\hat{\xi}}(t) = -K_2\tilde{q}(t) \end{cases} \qquad (17)$$

where $\hat{q}(t)$ and $\hat{\xi}(t)$ represent the estimates of $q(t)$ and $\xi(t)$. The observer matrices $\begin{bmatrix} K_1 & K_2 \end{bmatrix}^T = \begin{bmatrix} 2wI_3 & w^2I_3 \end{bmatrix}^T$ where $w$ is the observer bandwidth.

Defining $\tilde{q}(t) = \hat{q}(t) - q(t)$, and $\tilde{\xi}(t) = \hat{\xi}(t) - \xi(t)$ are the estimates. The dynamics of $\tilde{q}(t)$ and $\tilde{\xi}(t)$ can be written as

$$\begin{cases} \dot{\tilde{q}}(t) = -K_1\tilde{q}(t) + \tilde{\xi}(t) \\ \dot{\tilde{\xi}}(t) = -K_2\tilde{q}(t) - \dot{\xi}(t). \end{cases} \qquad (18)$$

Next, (18) can be rewritten as

$$\dot{E}_o(t) = TE_o(t) - D\dot{\xi}(t) \qquad (19)$$

where $E_o(t) = \begin{bmatrix} \tilde{q}^{\mathrm{T}}(t) & \tilde{\xi}^{\mathrm{T}}(t) \end{bmatrix}^{\mathrm{T}} \in \Re^6$ and

$$T = \begin{bmatrix} -K_1 & I_3 \\ -K_2 & 0_3 \end{bmatrix}, D = \begin{bmatrix} 0_3 \\ I_3 \end{bmatrix}.$$

Then, the following tracking error is defined as $e_1 = p - p_0(\theta)$. By taking the derivative of $e_1$ and using (6), we can get

$$\dot{e}_1 = q - p_0^\theta(\theta)\dot{\theta}. \tag{20}$$

Let $u_d - \vartheta(t) = \dot{\theta}(t)$, one can obtain

$$\dot{e}_1 = q - p_0^\theta(\theta)(u_d - \vartheta). \tag{21}$$

The kinematic guidance law $q_d$ is designed as follows to stabilize $e_1$:

$$q_d = -k_1 e_1 + p_0^\theta(\theta)u_d \tag{22}$$

and

$$\dot{\vartheta} = -\ell(\vartheta + \mu p_0^\theta(\theta)^{\mathrm{T}} e_1) \tag{23}$$

where $k_1 = \mathrm{diag}\{k_{11}, k_{12}, k_{13}\}$, $\ell$ and $\mu$ are positive constants.

To proceed, defining $e_2 = q - \hat{q}_d$, where $\hat{q}_d$ is the estimate of $q_d$. $\hat{q}_d$ can be obtained by using the following filtering scheme:

$$t_d \dot{\hat{q}}_d + \hat{q}_d = q_d, \quad \hat{q}_d(0) = q_d(0) \tag{24}$$

where $t_d$ is a positive constant. Let

$$e_d = \hat{q}_d - q_d. \tag{25}$$

And $\dot{q}_d \triangleq a = \begin{bmatrix} a_1 & a_2 & a_3 \end{bmatrix}^T$. $a_j$ is bounded by $|a_j| \leq a_j^*$, $j = 1, 2, 3$, where $a_j^*$ is a positive constant. For details, please refer to [26].

Then, the time derivative of $e_2$ yields

$$\dot{e}_2 = \xi + RM^{-1}\tau(t) + \frac{e_d}{t_d}. \tag{26}$$

To stabilize $e_2$, the anti-disturbance control law is developed as follows:

$$\tau_c(t) = MR^T(-\hat{\xi} - e_1 - \frac{e_d}{t_d} - k_2 e_2) \tag{27}$$

where $k_2 = \mathrm{diag}\{k_{21}, k_{22}, k_{23}\}$. Denote $\tau = \tau_c + \tau_e$.

### C. Safety-critical Obstacle Avoidance Controller

In this part, considering the collision with obstacles and ASV to design the optimal surge and sway force of safety conditions. From (8), we can get

$$\begin{aligned} \dot{\bar{p}} &= \bar{q} \\ \dot{\bar{q}} &= \hat{\xi}_2 + \tau_2 - \tilde{\xi}_2 \end{aligned} \tag{28}$$

where $\bar{p}$ denotes $[x(t), y(t)]^T$, $\bar{q}$ denotes $R_2(\psi)[u, v]^T$. $\hat{\xi}_2$, $\tilde{\xi}_2$ and $\tau_2$ is the first two dimensions of $\hat{\xi}$, $\tilde{\xi}$ and $\tau$, respectively. $\bar{p}_k = [x_k, y_k]^T$ is position of $k$th obstacle.

We choose the following candidate ISSf-ECBF

$$S_k(s) = \|\bar{p}_{ek}\|^2 - (r_k + d_k)^2 \tag{29}$$

where $\bar{p}_{ek} = \bar{p} - \bar{p}_k$, $s = [\bar{p}^T, \bar{q}^T]^T$. To achieve the objective of obstacle avoidance, the set $\mathcal{C}$ can be obtained

$$\mathcal{C} = \left\{ \bar{p} \in \mathbb{R}^2 : S_k(s) = \|\bar{p}_{ek}\|^2 - (r_k + d_k)^2 \geq 0 \right\} \tag{30}$$

For ease of notation, it is denoted by $S_k$ in the sequel. The safety constraint with $S_k(s)$ is described as

$$\mathcal{U} = \left\{ \tau_2 : \mathcal{L}_f^2 S_k + \mathcal{L}_g \mathcal{L}_f S_k \tau_2 - (\frac{\partial(\mathcal{L}_f S_k)}{\partial x})^T (\frac{\partial(\mathcal{L}_f S_k)}{\partial x}) + \mathcal{T}_s^T \mathcal{H}_s \geq 0 \right\} \tag{31}$$

where $\mathcal{L}_f^2 S_k = 2\bar{q}^T \bar{q} + 2\bar{p}_{ek}^T \hat{\xi}_2$, $\mathcal{L}_g \mathcal{L}_f S_k = 2\bar{p}_{ek}^T$, $\mathcal{T}_s = [\beta^2, 2\beta]^T$. For the ASV, ensuring safety takes precedence over geometric objectives. Based on the safety constraint (31), the following quadratic programming problem is constructed.

$$\tau^* = \underset{\tau \in \Re^m}{\mathrm{argmin}} J(\tau) = \|\tau - \tau_c\|^2$$

$$\text{s.t.} - \mathcal{L}_g \mathcal{L}_f S_k \tau \leq \phi \tag{32}$$

where $\phi = 2\bar{q}^T \bar{q} - -(\frac{\partial(\mathcal{L}_f S_k)}{\partial x})^T (\frac{\partial(\mathcal{L}_f S_k)}{\partial x}) + 2\bar{p}_{ek}^T \hat{\xi}_2 + \mathcal{T}_s^T \mathcal{H}_s$. The $\tau^*$ is obtained by solving the above quadratic programming problem.

**Remark 2.** *The proposed safety-critical controller can avoid obstacles while ensuring minimal impact on the given tracking task.*

## IV. STABILITY AND SAFETY ANALYSIS

In this section, we will conduct stability and safety analysis of the closed-loop system.

### A. Stability Analysis

**Lemma 2.** *The observer error subsystem in (19) is ISS, and the error signals being $\tilde{q}$ and $\tilde{f}$ are bounded by*

$$\|E_o(t)\| \leq \sqrt{\frac{\lambda_{\max}(N)}{\lambda_{\min}(N)}} \max\{\|E_o(t_0)\|e^{-\gamma_1(t-t_0)/2},$$

$$\frac{2\|ND\|\xi^*}{\varsigma_1 \kappa}\}, \forall t \geq t_0 \tag{33}$$

*where $\gamma_1 = ([\varsigma_1(1 - \kappa)]/[\lambda_{\max}(N)])$ and $0 < \kappa < 1$ provided that*

$$T^T N + NT \leq -\varsigma_1 I \tag{34}$$

*where $\varsigma_1 \in \Re$ is a positive constant.*

*Proof.* Choose the following Lyapunov function

$$V_1 = (1/2)E_o^{\mathrm{T}}(t)NE_o(t). \tag{35}$$

Taking (34) into account, one has $\dot{V}_1 = E_o(t)^{\mathrm{T}} N(TE_o(t) - D\dot{\xi}(t)) \leq -\frac{\varsigma_1}{2}\|E_o(t)\|^2 + \|E_o(t)\|\|ND\|\|\dot{\xi}(t)\|$ Since $\|E_o(t)\| \geq [(2\|ND\|\|\dot{\xi}(t)\|/\varsigma_1 \kappa)]$, we have

$$\dot{V}_1 \leq -\frac{\varsigma_1}{2}(1 - \kappa)\|E_o(t)\|^2. \tag{36}$$

It follows that the observer error subsystem described by (19) is ISS. It is important to note that $V_1$ is bounded and satisfies the inequality $([\lambda_{\min}(N)]/2)\|E_o(t)\|^2 \leq V_1 \leq ([\lambda_{\max}(N)]/2)\|E_o(t)\|^2$. From this, we can derive (33). □

Next, we will outline the stability analysis of the closed-loop system.

**Lemma 3.** *Taking into account the error dynamics represented by (21) and (26), the error signals $e_1$, $e_2$, $e_r$, $\vartheta$, and $\tilde{\gamma}$ are uniformly ultimately bounded by*

$$\|E\| \le \sqrt{\frac{\lambda_{\max}(Q)}{\lambda_{\min}(Q)}} \max\{\|E(t_0)\| e^{-\gamma_2(t-t_0)/2},$$

$$\frac{E_o + \|a^*\| + \varpi}{\epsilon \varsigma_2}\}, \forall t \ge t_0 \qquad (37)$$

*where $Q = diag\{1, 1/\ell\mu\}$, $\gamma_2 = 2\varsigma_2(1-\epsilon)/\lambda_{\max}(Q)$.*

*Proof.* The constructed Lyapunov function is

$$V_2 = \frac{1}{2}(e_1^T e_1 + e_2^T e_2 + e_d^T e_d) + \frac{\vartheta^2}{2\ell\mu}.$$

According to (20), (23)-(27), the time derivative of $V_2$ is

$$\dot{V}_2 = e_1^T(e_2 + e_d) - e_1^T k_1 e_1 + e_2^T f + e_2^T \frac{e_d}{t_d} + e_d^T(-\frac{e_d}{t_d} - a)$$

$$+ e_2^T(-\hat{f} - e_1 - \frac{e_d}{t_d} - k_2 e_2) + e_2^T RM^{-1}\tau_e - \frac{\vartheta^2}{\mu}. \quad (38)$$

Finally, we can obtain

$$\dot{V}_2 \le -(\lambda_{\min}(k_1) - \frac{1}{2})\|e_1\|^2 - (\frac{1}{t_d} - \frac{1}{2})\|e_d\|^2 + \|e_d\|\|a^*\|$$

$$- \lambda_{\min}(k_2)\|e_2\|^2 + \|e_2\|\|\tilde{f}\| + \|e_2^T\|\|M\tau_e\| - \frac{\vartheta^2}{\mu}. \quad (39)$$

Choose $\lambda_{\min}(k_1) - \frac{1}{2} > 0$, $\frac{1}{t_d} - \frac{1}{2} > 0$. Then, define $\varsigma_2 = \min(\lambda_{\min}(k_1) - \frac{1}{2}, \lambda_{\min}(k_2), \frac{1}{t_d} - \frac{1}{2}, \frac{1}{\mu})$, $\varpi = \|M\tau_e\|$, $E = \begin{bmatrix} e_1^T & e_2^T & e_d^T & \vartheta \end{bmatrix}^T$. Hence, (39) becomes $\dot{V}_2 \le -\varsigma_2\|\|E\|^2 + \|E\|(E_o + \|a^*\| + \varpi)$ and $\dot{V}_2 \le -\varsigma_2(1-\epsilon)\|E\|^2 + \|E\|(-\epsilon\varsigma_2\|E\| + E_o + \|a^*\| + \varpi)$ where $\epsilon \le 1$.

Note that

$$\|E\| \ge \frac{E_o + \|a^*\| + \varpi}{\epsilon \varsigma_2}$$

renders

$$\dot{V}_2 \le -\varsigma_2(1-\epsilon)\|E\|^2. \qquad (40)$$

It can be established that the error subsystem related to obstacle avoidance control is ISS. Additionally, the errors of the closed-loop system satisfies (37). $\square$

*B. Safe Analysis*

The subsequent lemma presents the safety analysis of the ASV.

**Lemma 4.** *Given the dynamics of the ASV as outlined in (6), if $\bar{p}(t_0) \in \mathcal{U}$ and $\bar{\tau}^* \in \mathcal{U}$ are satisfied for all $t > t_0$, the closed-loop system will be ISSf.*

*Proof.* According to Lemma 1, if $\bar{\tau}^* \in \mathcal{U}$ holds, then $\mathcal{C}$ is forward invariant, meaning that the set $\mathcal{C}$ is ISSf. In other words, as long as $\bar{p}(t_0)$ is within $\mathcal{C}$, the position $\bar{p}(t)$ will remain in $\mathcal{C}$ indefinitely. Therefore, the closed-loop system is ISSf. $\square$

**Theorem 2.** *The closed-loop system is shown to achieve ISSf, indicating that collision avoidance is feasible. Furthermore,*

all error signals in the closed-loop system are uniformly ultimately bounded.

*Proof.* According to Lemma 4, the ASV will meet the safety constraint, meaning that the safety objective (11) is fulfilled. We employ Lemmas 2, 3, and [27, Lemma 1], which enable us to deduce that the closed-loop system is ISSf. The norm $\|E\|$ is uniformly ultimately bounded by

$$\|E\| \le \sqrt{\frac{\lambda_{\max}(Q)}{\lambda_{\min}(Q)}}(\sqrt{\frac{\lambda_{\max}(N)}{\lambda_{\min}(N)}} \frac{2\|ND\|\xi^*}{\varsigma_1\kappa\epsilon\varsigma_2}$$

$$+ \frac{\|a^*\| + \varpi}{\epsilon\varsigma_2}). \qquad (41)$$

Given that $E$ is bounded, we can deduce that $e_1$ and $\vartheta$ are also bounded. As a result, it follows that $\|p(t) - p_0(\theta)\| = \|e_1\|$ and $\|\dot{\theta}(t) - u_d\|$ remain bounded, that is, (9) and (10) hold. $\square$

**Remark 3.**

## V. SIMULATION RESULTS

To validate the effectiveness of the proposed control strategy, this paper conducts simulations using Cybership II in [28]. The simulation parameters are set as follows: $w = 40$,

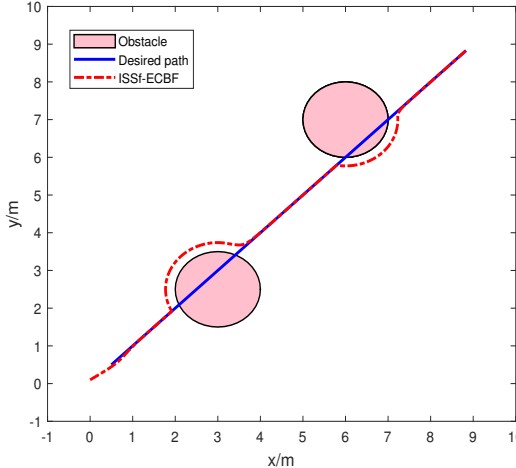

Fig. 2. Control performance of the proposed control strategy for ASV.

$\Omega = 0.1$, $l = 1$, $\mu = 0.1$, $\beta = 30$, $d_k = 0.5$, $r_k = 1$, $k_1 = \text{diag}\{3, 2, 8\}$, $k_2 = \text{diag}\{16, 22, 28\}$, $t_d = 0.1$, $d_w = \begin{bmatrix} 3\cos(t)\sin(0.5t) & 4\sin(0.5t)\cos(0.5t) & 0.2\sin(t) \end{bmatrix}^T$. The desired parameterized path is $x_d(\vartheta_0) = y_d(\vartheta_0) = 0.06\vartheta_0 + 0.5$, $\psi_d = \pi/4$. The position of static obstacle is $\bar{p}_1 = \begin{bmatrix} 3 & 2.5 \end{bmatrix}^T$, $\bar{p}_2 = \begin{bmatrix} 6 & 7 \end{bmatrix}^T$.

Fig. 2 illustrates the effectiveness of the safety-critical controller proposed for the autonomous vessel. The upper section demonstrates that the vessel prioritizes obstacle avoidance to ensure safety. Once the obstacle avoidance operation is complete, it can proceed with the tracking task. Fig. 3 illustrates the observation effect of the extended state observer. The velocity and aggregated disturbances of the ASV can be

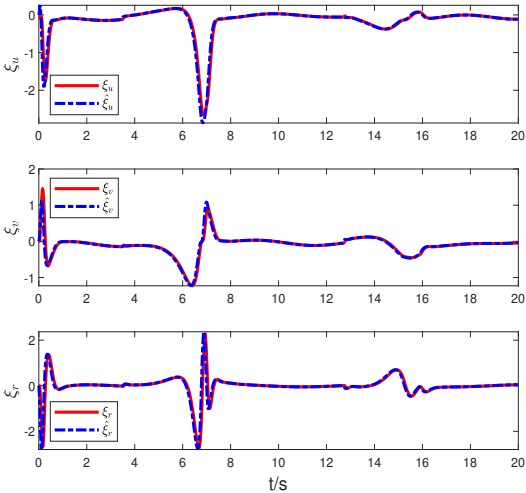

Fig. 3. The estimates of lumped disturbances.

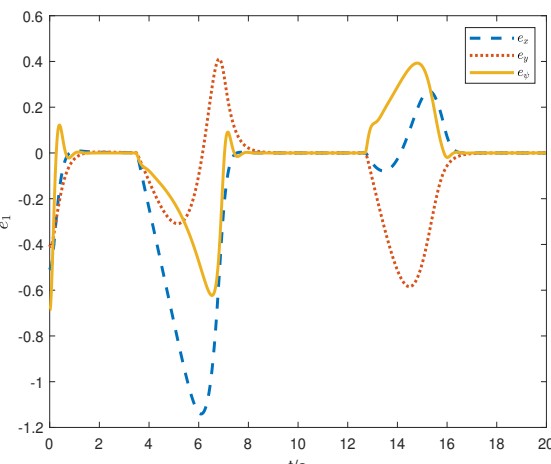

Fig. 4. Tracking errors of the ASV.

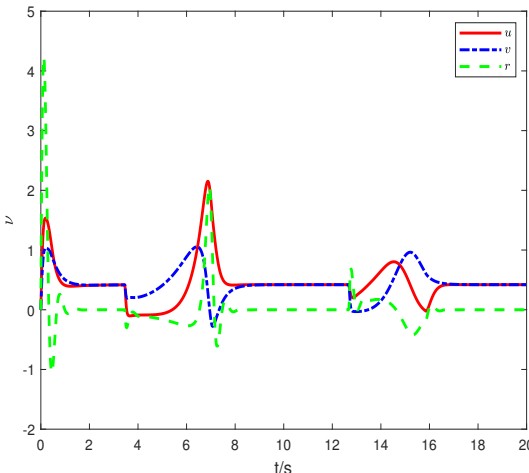

Fig. 5. Velocity comparisons of the ASV.

accurately estimated. Fig. 4 and Fig. 5 depict the comparisons of the tracking errors and velocities, respectively.

## VI. Conclusion

This paper introduces a safety-critical control strategy for ASVs that considers external marine disturbances and internal model uncertainties. Initially, an anti-disturbance controller is devised based on the estimation of lumped disturbances using an ESO. Following this, a quadratic optimization problem is established by incorporating ISSf-ECBFs to enforce safety constraints on the control inputs. By solving this problem, a safety-critical controller is derived, significantly improving the safety and robustness of the system. The closed-loop system is demonstrated to be ISSf, with error signals shown to be uniformly ultimately bounded. The effectiveness of the proposed control strategy is verified through simulation results.

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
