# OpenReview forum: "Safety-critical Obstacle Avoidance Control of Autonomous Surface Vehicles with Uncertainties and Disturbances"
_IEEE.org/ICIST/2024/Conference — IEEE ICIST 2024 Conference Submission_

### Official Review · Reviewer_XW49 · 2024-08-21
**Accept**

**Rating:** 7
**Confidence:** 4

**Review:**

This paper presents a safety-critical obstacle avoidance control strategy for autonomous surface vehicles, addressing external marine disturbances and internal model uncertainties. It extends the Input-State Safety Index Control Barrier Function to handle unknown disturbances and designs an extended state observer for estimating lumped disturbances. Based on the observer's results, a disturbance rejection controller is designed. The closed-loop system's input-state safety and the boundedness of the error signals are proven, and the effectiveness of the control strategy is validated through simulations.

1.How accurately can the unknown lumped disturbances be estimated in practice?

2.What is the computational efficiency of the quadratic programming problem, and is it suitable for real-time control applications?

3.How robust is the control strategy against model uncertainties and disturbances in various marine environments?

4.Are there more concise or efficient control strategies that can meet the requirements for safety-critical obstacle avoidance?

---

### Official Review · Reviewer_77B5 · 2024-08-21
**This paper is well structured, logical and of practical value and is recommended for publication.**

**Rating:** 7
**Confidence:** 4

**Review:**

The paper introduces a significant advancement in the field of autonomous surface vehicles (ASVs) by addressing safety-critical obstacle avoidance in the presence of disturbances and uncertainties. The extension of exponential control barrier functions (ECBFs) to account for unknown disturbances through the development of input-to-state safe exponential control barrier functions (ISSf-ECBFs) represents a noteworthy contribution. This enhancement ensures that ASVs can maintain safety even when faced with unpredictable external conditions, a critical requirement for real-world applications. The reviewer has the following questions to discuss with the authors:

1. How well does the proposed ISSf-ECBFs approach perform in real-world scenarios where disturbances might be more complex or varied than those simulated? Are there plans for testing the system in real maritime environments?

2. The paper relies on an extended state observer to estimate unknown external disturbances and internal uncertainties. How sensitive is the overall control strategy to inaccuracies in these estimations? What happens if the observer’s estimations are significantly off?

3. While the paper addresses unknown disturbances, how does the control strategy perform under extreme or highly irregular disturbances that might exceed typical operational conditions? Is there a threshold beyond which the safety guarantees no longer hold?

4. Over extended periods of operation, how does the control strategy account for potential changes in the performance of ASVs or wear-and-tear that could affect the accuracy of the state observer and overall system safety?

---

### Official Review · Reviewer_JYue · 2024-08-22
**This paper proposes a safety-critical obstacle avoidance control approach for autonomous surface vehicles (ASVs) with disturbances and uncertainties. The feasibility of the designed control approach is proven via the simulation example. However, the following suggestions need to be considered in the revised manuscript to further improve the quality of this paper.**

**Rating:** 7
**Confidence:** 3

**Review:**

1. How does the extension of ISSf-ECBFs to handle unknown disturbances improve safety constraints compared to the method in [21]?
2. How does the safety-critical controller formulated in this paper using ISSf-ECBFs differ from the approaches in [12], [22]-[24]?
3. What are the main benefits of using a quadratic programming problem for collision avoidance with obstacles?

---

### Decision · Program_Chairs · 2024-09-08

Accept (Oral)